# Dietary Intake Patterns and Lifestyle Behaviors of Pregnant Women Living in a Manitoba First Nations Community: Implications for Fetal Alcohol Spectrum Disorder

**DOI:** 10.3390/nu14153233

**Published:** 2022-08-07

**Authors:** Olena Kloss, Marie Jebb, Linda Chartrand, Albert E. Chudley, Michael N. A. Eskin, Miyoung Suh

**Affiliations:** 1Department of Food and Human Nutritional Sciences, University of Manitoba, Winnipeg, MB R3T 2N2, Canada; 2Division of Neurodegenerative Disorders, St. Boniface Hospital Albrechtsen Research Centre, Winnipeg, MB R2H 2A6, Canada; 3Canadian Centre for Agri-Food Research in Health and Medicine, St. Boniface Hospital Albrechtsen Research Centre, Winnipeg, MB R2H 2A6, Canada; 4Beatrice Wilson Health Centre, Opaskwayak Cree Nation, Opaskwayak, MB R0B 2J0, Canada; 5Department of Pediatrics and Child Health, University of Manitoba, Winnipeg, MB R3A 1S1, Canada; 6Department of Biochemistry and Medical Genetics, Max Rady College of Medicine, University of Manitoba, Winnipeg, MB R3A 0A2, Canada

**Keywords:** alcohol, FASD, First Nations, macronutrients, maternal nutrition, micronutrients

## Abstract

The information on the nutrition status of women at-risk of carrying a child with fetal alcohol spectrum disorder (FASD) is scarce, particularly in the First Nations population living on reserve. This study examined and compared nutrition status, dietary intake, and lifestyle patterns of pregnant at-risk, defined as those who consume alcoholic drink during the current pregnancy, and non-at-risk women living in northern Manitoban community. Thirty-seven pregnant, First Nations women (at-risk *n* = 15; non-at-risk, *n* = 22) were recruited to participate in the study. A questionnaire, presented in paper and iPad formats, collected information on participants’ demographics, dietary intake, lifestyle, pregnancy outcomes, and maternal health. A food frequency questionnaire and 24-h recall were used to determine nutrient intake. Nutrient values were assessed using Dietary Reference Intakes (DRI). At-risk and non-at-risk women were below the Canada Food Guide serving size recommended for Vegetable and Fruit, Grain, and Milk Products with 93%, 92%, and 93% of participants not meeting the recommendations, respectively. Women met the recommendations for vitamins A, B1, B12, C, niacin, choline, as well as calcium, and zinc. Sixty eight percentage (%) of participants did not meet the recommendations for folate and iron, and 97% for docosahexaenoic acid (DHA). Significant differences were observed between non-at-risk and at-risk women for mean % DRI intakes of vitamin C (313 ± 224 vs. 172 ± 81 mg/day), niacin (281 ± 123 vs. 198 ± 80 mg/day), folate (70 ± 38 vs. 10 ± 22 mcg/day), and iron (101 ± 74 vs. 74 ± 30 mg/day). The findings of this study lay a fundamental premise for the development of community nutrition programs, nutrition education, and nutrition intervention, such as community specific prenatal supplementation. These will assist in ensuring adequate maternal nutrient intake and benefit families and communities in Northern Manitoba with and without alcohol insult.

## 1. Introduction 

Nutrient intake and dietary patterns during gestation are stated to be the most critical modifiable factors shaping pregnancy and birth outcomes. Meeting dietary recommendations for macro- and micro-nutrients during gestation is necessary to sustain the needs of a developing fetus and a mother. The state of gestation increases caloric requirements by 13%, protein requirements by over 50%, and vitamin and mineral requirements by 0–50%, depending on the micro-nutrient [1]. The consequences of inadequate energy and nutrient intake are well documented [2,3]. 

Substance use during gestation, specifically alcohol (in the form of ethanol, EtOH), presents a burden on maternal dietary demands as EtOH influences macro and micro-nutrient status. Although EtOH provides 7 kcal/g, it does not have any nutritional value [4]. EtOH compromises nutrition status through primary and secondary malnutrition, by displacing normal caloric intake with EtOH-derived calories and impairing gastrointestinal function resulting in impaired transport and metabolism [4,5]. It has been proposed that compromised nutrition status coupled with EtOH intake during gestation leads to a severe form of fetal alcohol spectrum disorder (FASD)-terminology used for a spectrum of permanent disabilities resulting from gestational EtOH exposure [6,7]. 

Although the true prevalence of prenatal alcohol exposure (PAE) is not known, the Canadian Maternity Experiences Survey reports that 10.8% of Canadian women consumed EtOH while pregnant [8]. It has been reported that PAE might be higher in Indigenous communities [9]. Manitoba First Nations Regional Longitudinal Health Survey (RHS) indicated that 15% of surveyed women continued EtOH consumption after learning about their pregnancy [9]. A recent report on avoidable mortality in First Nations women demonstrated that the risk of death due to substance use, either EtOH or an illicit drug, was ten times higher than for the non-Indigenous population [10]. Higher rates of EtOH use often translate into increased rates of FASD. 

Maternal health disparity among First Nations women is further marked by adverse nutrition-related factors such as food insecurity. According to the First Nations RHS (2012), 54.2% of on-reserve households are food insecure [11]. In Manitoba, household food insecurity increases as the latitude increases, reaching astounding rates of 100% north of the 50th parallel [12]. Precarious access to financial resources, food, and goods creates an unfavorable dietary environment for an expecting mother and a child, leading to a multitude of negative health consequences such as low birth weights [13], damage to fetal development [14], maternal and fetal malnutrition [3], obesity [15], and intrauterine growth restriction [16]. Poor access to resources coupled with substance use can aggravate negative impacts on maternal and child health outcomes, leading to severe irreversible developmental damages [6,12]. 

While the impact of nutrition on maternal and fetal health is well documented, the nutrient intake and dietary pattern of First Nations pregnant women living on-reserve are largely unidentified. Furthermore, there is no information on dietary patterns and nutrient intake of First Nation women at-risk of PAE. The lack of such information obstructs appropriate maternal and child health program planning, restricts evidence-based policy development, and circumscribes FASD intervention and care provision. Therefore, the objective of this study is two-fold: (1) To identify the nutrient intake and dietary patterns, and maternal health of pregnant First Nations women living on a reserve; (2) to compare the above-stated based on risk behavior, specifically alcohol intake, during pregnancy. 

## 2. Experimental Design and Methodology

The study protocol was developed in consonance with the Health Research Involving Aboriginal Peoples Guidelines established by the Canadian Institutes of Health Research (CIHR) and the Tri-Council Policy (TCP) Statement, “Ethical Conduct for Research Involving Humans” [17]. The study protocol and consent forms were approved by the University of Manitoba Health Research Ethics Board (HS16448), as per Tri-Council Policy on Ethical Conduct for Research Involving Humans requirement. All data operation processes were conformed to Indigenous research principles of ownership, control, access, and possession (OCAP) [18]. The detailed process of protocol development is described previously as this study is the sub-part of a larger Manitoban cross-sectional study, investigating dietary risk and lifestyle risk factors for FASD [19]. 

Upon the receiving of the Band and Council Resolution (BCR), a community engagement process was initiated, which is located 7 h up north from Winnipeg. The process of community engagement focused on building relationships with the community members, elders, band and council, and health care team of the community’s Health Center. Involvement with the health center included development and the facilitation of maternal health workshops and delivery of the workshops at the community’s Prenatal Program. Pregnant women were recruited from this program through convenience sampling. Women who meet the inclusion criteria and expressed interest in the participation were provided with an information brochure, infograph about the study, and the consent form. Those individuals under 18 were provided a consent form for a caregiver and/or parents. In accordance with the trauma-informed practice, potential recruits were notified of the confidentiality and anonymity; they were also informed that if at any point they felt uncomfortable with the questions they could withdraw from participating. The participants were divided into two groups, at-risk group and non-at-risk group for the study purpose. At-risk group was defined as those who responded “yes” to the question “Did you have an alcoholic drink?” with respect to the current pregnancy, regardless of reported frequency of consumption.

The in-depth questionnaire, titled Nutrition for Two was utilized for the data collection process in this community. Two versions of the questionnaire were developed; a paper and an iPad version, created by a software developing company-Function Four Ltd. (Winnipeg, MB, Canada). The development of the questionnaire has been undertaken in consultation with several stakeholders, and professionals engaged with maternal programs in Winnipeg and First Nations communities across Manitoba, Canada. A detailed description of the construct definitions, dietary instruments development, and the approach to the analysis of the dietary-intake data is available [19]. Concisely, the questionnaire was composed of four sections:

(i) Demographics: included questions on demographics, such as age, marital status, ethnicity, place of residence. 

(ii) Dietary assessment: this section consisted of questions on daily meal patterns, also included two nutrition-intake assessment instruments: food frequency questionnaire (FFQ) developed based on Eating Well with Canada’s Food Guide (2007, CFG) [20] and Canadian Nutrient File [21]; and 24-h food recall. The FFQ was created to collect nutrient intake information of nutrients identified as important for pregnancy and fetal central nervous system development, which was based on an extensive literature review undertaken in our laboratory [22]. The list of food items listed is at least the top 10 sources of nutrients included: vitamin A, B1, B12, C, niacin, folate, choline, calcium, iron, zinc, and docosahexaenoic acid (DHA) [19,20]. The information on the quantity of a micronutrient in a given portion size came from the Canadian Nutrient File. Commonly consumed food items were based on the Canadian Community Health Survey (CCHS). Intake of First Nations traditional foods was also included. A total of 104 food items were selected for this instrument. The frequency of the consumption and portion sizes were derived from a previously validated questionnaire [23]. Although, a new CFG has been released in 2019, at the time of data collection and dietary analysis for this project former Canada’s recommendations were still in place. Therefore, the analysis and interpretation are constructed based upon former recommendations. The 24-h dietary recall requested detailed information on each food item and beverage consumed from the time a participant woke up till the time they went to bed, for the day prior to an interview. 

(iii) Maternal health: this section collected data on participants’ health status, medication and supplement use, anthropometric measurements. This section also asked questions on alcohol consumption, smoking, and drug use. 

(iv) Pregnancy outcomes: section comprised of questions on certain maternal characteristics, such as parity, gravidity, number of pre-term births, and bed rest. 

The statistical analysis was performed for the whole population as well as for women defined as at-risk and non-at-risk. The data are presented as all women, not-at-risk, and at-risk women. The non-exposed group was defined as those who answered “no”. All statistical analyses were conducted using IBM Statistics SPSS 26 (IBM Corp., New York, NY, USA). The significance level was set at 2-tailed alpha ≤ 0.05. All data were tested for normal distribution using Shapiro–Wilks and Kolmogorov–Smirnov tests. Differences for mean intakes between non-at-risk and at-risk groups were assessed by an independent *t*-test for normally distributed data. A non-parametric Wilcoxon rank-sum test was used for categorical and not normally distributed data. To test for the associations between categorical variables, 2 × 2 tables with the Fisher’s exact test were used, and for data with more than two categories, the Chi-square statistic was used. The odds ratio measured the association between PAE and not meeting the recommendations for food groups and micro-nutrients. Data were presented as mean ± standard deviation (SD) as well as median and (interquartile range, IQR), or percentage/proportion (%) of participants. 

## 3. Results

### 3.1. Basic Demographics

Sociodemographic and maternal health characteristics of women who were defined as non-at-risk and at-risk are presented in Table 1. Thirty-eight (38) women participated in this study but only 37 were included for data analysis since one participant did not provide dietary records. All participants were First Nations status women residing on the reserve with an average age of 24.4 ± 7.0 years old. The majority of women have reported completion of junior high and high schools (44% and 41%, respectively); and 39% reported unemployment and 28% student status. About half of the participants reported being on social assistance (51%). 

The average maternal pre-pregnancy body mass index (BMI) was estimated to be 26.5 ± 8.8 kg/m^2^, which is classified as an overweight status. Chronic illness before and during pregnancy was reported by 15% and 21%, respectively; with major chronic illnesses being type 2 diabetes and gestational diabetes. About 37% of participants reported smoking cigarettes during the given pregnancy, as well as 19% reported using illicit drugs such as marijuana (at the time of the data collection marijuana was illicit) and cocaine. There were no significant differences between non-at-risk and at-risk women with respect to any of the characteristics, except for age, with the at-risk women having higher average age (*p* < 0.05). 

### 3.2. Food Group Intake 

Table 2 presents food group intake for all participants, at-risk, non-at-risk groups, and reference values adjusted for dietary intake during pregnancy. The average Vegetable and Fruit and Grain Products food groups consumption was about 55 ± 33% of the recommended food group servings. The Milk and Alternative and Meat and Alternative food groups consumption was 67 ± 38% and 150 ± 100% of the food groups, respectively. There were no differences detected between non-at-risk and at-risk groups. 

A great number of participants were not meeting the recommendations. Vegetables and Fruit and Milk and Alternatives food groups were not met by 94% of participants; followed by Grain Products, where 89% of participants had not met the recommendations. The food group in which the least number of participants (86%) had not met the recommendations was Meat and Alternatives (Figure 1). There were no significant differences between at-risk and non-at-risk proportions for any of the four food groups. 

### 3.3. Macronutrient and Energy Intake 

The intake of each macronutrient is presented in absolute amount (gram/day, Table 3) and their energy distribution (%, total energy, Figure 2). While no difference was identified for the protein, carbohydrate, and sugar intake, women in the at-risk group had significantly (*p* < 0.01) higher daily fat intake (176%) compared to the women in the non-at-risk. This contributed to a higher total caloric intake in the at-risk-group (2342 ± 812 vs. 1807 ± 715 kcals/day, *p* < 0.05). The sugar alone intake was similar to protein intake, taking about 14% of energy contribution (Table 3). 

When deriving percent energy from macronutrients, this cohort’s energy intake was within the recommended reference values Acceptable Macronutrient Distribution Range (AMDR) for caloric intakes stemming from protein (17 ± 4%; AMDR 10–35%), carbohydrates (51 ± 9%; AMDR 45–65%) and fat (33 ± 8%; AMDR 20–35%) (Figure 2a). Congruently with macronutrient intake, energy intake from fat was significantly higher at the at-risk group than non-at-risk group, having 37 ± 8% and 29 ± 7% (*p* < 0.05), respectively (Figure 2b). The energy intake from protein and carbohydrates was within the recommended range for both at-risk and non-at-risk groups.

### 3.4. Micronutrient Intake

Daily intake of micronutrients is presented in absolute amount in Table 4. On average, all women were meeting and exceeding the Dietary Reference Intakes (DRI) recommendations for vitamin A, vitamin C, thiamin, niacin, vitamin B12, choline, calcium, and zinc. However, the intake of folate, iron, and DHA were below the recommendations, 86%, 88%, and 39%, respectively. When compared between the two groups, the daily intakes of vitamin C, niacin, folate and iron were significantly different. Women in the at-risk group, compared to non-at-risk group, had significantly lower intake of vitamin C (*p* < 0.02), niacin (*p* < 0.03), folate (*p* < 0.01) and iron (*p* < 0.04). Zinc intake was also lower in women at-risk group at the level of *p* < 0.06. 

Figure 3a,b depicts proportions and 95% confidence intervals (CIs) for the proportion of participants not meeting DRIs for all studied nutrients for all women and by alcohol exposure. The majority of participants met the recommendations for vitamin A, C, B1, B12, niacin, and zinc. However, folate, choline, calcium, iron, and DHA were not met by 68%, 45%, 58%, 68%, and 97% of participants (Figure 3a). A higher proportion of participants were not meeting the DRI recommendations in the at-risk group for vitamin C, vitamin B1, folate, choline, calcium, iron, and zinc. Although the statistically significant difference between the groups was detected only for folate (Figure 3b). 

## 4. Discussion

### 4.1. Demographics and Maternal Health Status

Demographic information demonstrates that the participating cohort is relatively homogenous with all women reporting First Nations Status and residence on the reserve. The average age was 24.4 ± 7.0 years old, suggesting that the age of first-time motherhood is below the average maternal age of the general Canadian population (29.6 years of age) [8]. This is consistent with the findings of other studies, pointing toward high fertility rates among First Nations teenage girls [25,26]. Maternal pre-pregnancy BMI was estimated to be overweight (26.5 ± 8.8 kg/ m^2^) with 33% being within normal BMI ranges. Chronic illness before and during pregnancy was increased from 15% to 21%, with the majority being type 2 diabetes and gestational diabetes. These findings are congruent with other studies which reported pre-pregnancy BMI of 28.3 ± 6.0 kg/m^2^, 25.3 ± 6.0 kg/m^2^, and 26.0 ± 6.0 kg/m^2^ respectively [25,27,28]. Conversely to the above-mentioned reports and national statistics on the proportion of pregnant First Nations women diagnosed with chronic illnesses (4.7%), this cohort had a notably higher proportion of diagnosed participants [8]. This is a concerning finding as the literature unequivocally points to an increased risk of pregnancy complications, namely gestational diabetes, infant macrosomia, preeclampsia [29] as well as infant co-morbidities such as childhood obesity and type 2 diabetes [30]. Increasing number of reports indicate that metabolic conditions and diets that lack appropriate nutrients during pregnancy have critical impacts on fetal programming and epigenetic modifications, which result in positive or negative health outcomes for an individual during early life and adulthood [29,30,31]. Thus, the maternal health programming needs to target women prior to conception, which proactively educate them on the benefits of healthy lifestyle, nutrition, and exercise. 

About 37% of participants reported smoking cigarettes, which is almost identical to the findings of Oliveira et al. of 35%, which is substantially higher compared to the general Canadian population (7%) [25]. Nineteen (19%) percent of participants reported using illicit drug use such as marijuana (at the time of the data collection marijuana was illicit) and cocaine. This is markedly higher than the general Canadian average of 1% [8]. Smoking and drug use during pregnancy exacerbate the teratogenic effects of alcohol, elicit fetal hypoxia and contribute to aberrations in nutrient metabolism, thus compounding the risk factor effects for FASD [32]. 

### 4.2. Food Group and Macronutrient Intake 

The analysis of food group intake presented provides an understanding of maternal food group intakes for First Nations women residing in the community. Women consumed all food categories but were not meeting the CFG recommendations for Vegetable and Fruit and Grain Products. Although the average intake of recommended servings in CFG was over 50% and higher for all the food groups, a great number of participants were not meeting the recommendations. Although little comparative information is available with respect to maternal food group intake, the findings of this study are consistent with Johnson-Dawn and Egeland’s study (2013), where First Nations women of child-bearing age did not meet the recommendations Fruits and Vegetables (>90% of participants), Grain Products (>70% of participants) and Milk and Alternatives (>90% of participants) [33]. It is concerning that the cohort consisting of expecting women with higher incidences of obesity had low intakes of nutrient-dense and fiber-rich foods such as fruits, vegetables, dairy, and whole-grain products. Lower intakes of these food groups could be due to the nutrition transition experienced by Indigenous people in remote communities. Researchers argue that nutrition transition diverted Indigenous people away from traditional foods and has led to the adoption of foods from non-indigenous sources, which are higher in calories and undergo processing [34].

The macronutrient intake for all participants from all food sources was very similar to the intake cited by Back and colleagues who investigated dietary intake and physical activity in First Nations pregnant women living in rural and urban settings [35]. The study reported 308.8 ± 121.0 g/day coming from carbohydrates, 83.1 ± 31.8 g/day from protein, and 81.2 ± 52.5 g/day coming from fat. These intakes corresponded with the intakes of the participants in this study. No Canadian comparative literature is available on maternal macronutrient intake for women at-risk. Regardless a close monitoring of total energy intake is important since poorer energy intake is related with the maternal weight gains and neonatal birth outcomes in both human and animal studies [36,37].

### 4.3. Micronutrient Intake 

Although there is huge variation among participants, results for micronutrient intakes reveal that the community’s pregnant women are meeting and exceeding the DRI recommendations for vitamins A, C, B1, B12, niacin, choline, calcium, and zinc. However, the intake of folate, iron, and DHA was below that of DRI recommendations. In addition, the findings indicate that micronutrients such as folate, choline, calcium, iron, and DHA were not met by 68%, 45%, 58%, 68%, and 97% of participants, respectively. Similar micronutrient inadequacy for some of the afore-stated micronutrients was displayed by other Canadian studies. A report by Berti and colleagues revealed that pregnant, lactating, and women of child-bearing age, residing in Canadian Arctic communities had inadequate intake levels of magnesium, calcium, vitamins A, C, E, and folate as well as infrequent use of the nutritional supplement, which was also identified in the cohort [38]. A Manitoban report by Chan and colleagues corroborated these findings by detecting inadequate intakes for vitamin A, folate, calcium, and iron. However, this report did not focus on pregnant and lactating women [39]. 

The sub-analysis of non-at-risk and at-risk groups revealed that the daily intakes of vitamin C, niacin, folate, and iron were significantly lower among the pregnant women at risk, with 93% of women having an inadequate intake of folate. A very recent prospective birth cohort study [36] reported that gestational weight gain and inadequate prenatal intakes of iron and choline in combination with PAE resulted in suboptimal growth of the infants measured at 2 week and 12 months. Unfortunately, the authors’ findings cannot be confirmed in our present study, since no birth outcome data were obtained, which warrants the next phase study. No other known Canadian studies have reported on the differences in the intakes of micronutrients based on EtOH consumption during gestation. The finding of low folate intake in the at-risk group, as well as the overall cohort, is likely due to low intake of dark green leafy vegetables, beans, and grains which are good folate sources. This finding is concerning due to the crucial role of folate in DNA methylation, cell differentiation, and proliferation, which occurs at a higher rate in the first 12 weeks of gestation. In the presence of EtOH intake, folate metabolism is further complicated through reduced uptake of folate from jejunum, decreased hepatic storage [40], and increased urinary excretion [41]. This makes alcohol-consuming individuals particularly vulnerable to an already existing folate deficiency. 

Women in this present cohort did not achieve the Food and Agriculture Organization (FAO) recommendations of DHA 200 mg/day for pregnant women [42]. The average intake of the entire cohort was only 78 mg/day regardless of EtOH consumption, which is substantially lower than the recommended intake. Similar findings have been reported by other Canadian studies. Denomme et al. reported an average intake of 82 ± 33 mg/day with only 10% of participants meeting the recommendations [43]. A study of pregnant women in British Columbia (BC) reported a mean DHA intake of 146 ± 161 mg/day which is markedly higher than the intake in our cohort [44]. Congruous with our study, the BC study reported a high degree of variability in DHA dietary intake, equating to over 2000% [44]. Another Canadian study, which compared DHA and fish intake of pregnant women to FAO/World Health Organization (WHO) recommendations revealed a substantially higher intake of DHA (237 ± 164 mg/day) compared to our cohort [45]. Furthermore, the recommendation for DHA intake was met by 53% of women compared to the 3% of women in our study. However, the participating women in this study were from a coastal area with higher fish supply, high socioeconomic status, and high educational attainment which rendered an increased likelihood of participants’ awareness of the benefits of omega-3 fatty acids intake [45]. While the afore-mentioned study recruited pregnant participants, small sample sizes (*n* = 54) and lack of Indigenous participants do not allow for valid comparisons [45]. 

Although the compounding effects of EtOH and macro- and micro-nutrient deficiencies on the embryogenic and fetal development have not been elucidated, studies indicate that EtOH-induced insults are aggravated by poor nutrient intake and nutrient status [22]. A number of experimental reports link poor intake of nutrients with PAE to microcephaly, intrauterine growth retardation (IUGR), abnormalities in cardiogenesis, and severe valve malfunctions [22,46]. In light of this information, the findings of low folate, DHA, and iron intake in the risk group raise special concerns.

Health disparities are shaped by structural inequities that formed as a result of historic policies targeted at assimilation of the Indigenous population [47,48]. The history of colonization and the residential school system lay at the foundation of inequities in the health and well-being of the Indigenous population [48]. The historic and social analysis provides evidence on how cumulative effects of colonial infringements engendered loss of identity, resources, agency, and language, culture, and land. Specifically, in the gendered dimension, the colonial infringements greatly impacted Indigenous women’s autonomy, motherhood, and feminine capacity [49,50]. Despite the existence of data suggesting the feasibility of nutrition program as a potential prevention approach for FASD, at present there are no programs in Canada that utilize nutrition intervention as a prevention method for FASD [51]. Formidable factors in implementation of nutrition intervention as FASD prevention strategy are lack of clinical trials in human participants and lack of information on maternal nutrition status who are at-risk [51]. Therefore, studies that examine maternal nutritional status need to be conducted in order to establish the baseline nutrition status of pregnant women prior to making the recommendations for nutrition supplementation provisions. This project may serve as a great milestone to developing sound nutrition intervention for FASD in at-risk populations.

### 4.4. Strengths 

One particular strength of this study lies in the community’s involvement in the planning and execution of the project. Extensive community consultations were performed at all stages of the project. This greatly contributed to building a trust-based relationship with the community residents, Beatrice Wilson Health Centre staff, elders, and participants. Another noteworthy strength of the study is the continuous and consistent in-person presence of the researcher to have face-to-face meetings in the community, taking 14 h of driving time for each visit. Additionally, a community worker was hired to assist with recruitment and relationship building with the community. This allowed the team to establish a trustworthy relationship with the participants. Studies indicate that a collaborative and involved approach to research with Indigenous communities increases the probability of knowledge transfer, which occurs during the research process [50]. Furthermore, collaborative research creates distinctive opportunities for a researcher to take on a more profound perspective of the community and participants, thus enabling a more critical evaluation of results and interpretation [48]. 

### 4.5. Limitations

While this is a new initiative, one of the most critical limitations of this project was a small sample size. Due to the distance from the city of Winnipeg (630 km) to the rural community, the ability to collect an appropriate number of participants was constricted. Although the present study is a pilot study and sample size calculations were not performed, the sample size of 37 impeded various forms of sub-analysis, such as comparisons of macro-and micro-intakes between the three trimesters. This sample size may be representative of the small maternal population of the community. According to the Census, the total number of community citizens of all ages is approximately 2500 [52]. However, this sample size may generate complications for extending these findings to the larger on-reserve maternal population. There is lack of comparable studies not only for Canadian First Nations population, but general Canadian population. Although there are a few reports identifying nutrient intake of First Nations women at various stages of reproductive phases, at the present time, there are no reports identifying intakes for pregnant women at-risk either at the community, provincial, or federal level. A small number of international reports, which examine various dietary components with at-risk sub-analysis, have similarly smaller sample sizes ranging from 10 to 33 individuals in the at-risk group [53,54]. Therefore, this study provides the expected prevalence—a crucial variable in the calculation of the sample size for cross sectional designs. This study’s results could further be utilized for similar cross-sectional designs, which aim to estimate dietary intake for First Nations maternal populations. 

Another limitation of the project is the use of self-reported data, which generates participant’s response bias and complicates accurate assessment and interpretation. One major criticism of self-reported data is the extent of error in portion size reporting [55]. This error can be substantial, as supported by steady findings from comparisons studies of self-reported total energy expenditure [55]. To mitigate the impact of this limitation on the study results food models were used, as well as re-affirming and clarifying questions with respect to portion sizes, brand names, and frequency of intake were asked. 

The collection of sensitive information induced limitations to transparent data collection. Not all participants answered the questions about EtOH intake, drug use, and smoking during pregnancy. Although re-assured of confidentiality, some women decided not to disclose their answers. This might have been due to the intrusiveness factor and respondents might have viewed it as an invasion of privacy and inappropriate for the conversation. Another factor contributing to the lack of responses could have involved social desirability, as consumption of alcohol during pregnancy is generally deemed as socially undesirable behavior. 

Lack of extensive validity and reliability testing posed another set of limitations for this project. Although the questions of the survey instrument were derived from CCHS, and intensively reviewed by the FASD First Nation community coordinators, and content validity was performed through Mothering Project at Mount Carmel Clinic (Winnipeg, MB, Canada) [19], the survey instrument did not undergo reliability tests. Testing for reliability is pivotal as it yields information about congruency and consistency of the instrument items, across the instrument. 

## 5. Conclusions

Nutrition intervention during pregnancy is a promising FASD prevention strategy; however, it is overlooked due to lack of data on nutritional status. The findings presented in this study provide important baseline data on dietary patterns and nutrient intake of macro- and micronutrients important for fetal central nervous system (CNS) development of First Nations pregnant women residing in the northern Manitoban reserve. It also identifies differences in dietary patterns and micronutrient intakes between women who are at-risk and non-at-risk of carrying a child with FASD. Future research should focus on exploring in more detail the patterns of maternal health with respect to biological nutrition markers, EtOH markers, and neonatal outcomes. Strengthening nutrition research methodologies for the premise of FASD programming will positively impact all not only maternal and child health outcomes, but also build awareness around the significance of nutrition as a determinant of maternal and child health. Furthermore, studies should further investigate the relationships between the timing of pregnancy, the amount and timing of alcohol consumption during gestation, and the duration of ethanol exposure. The information presented in this study should be utilized by health professionals, dietitians, clinicians, and public health professionals for the planning of preventative maternal health care delivery on reserve.

## Figures and Tables

**Figure 1 nutrients-14-03233-f001:**
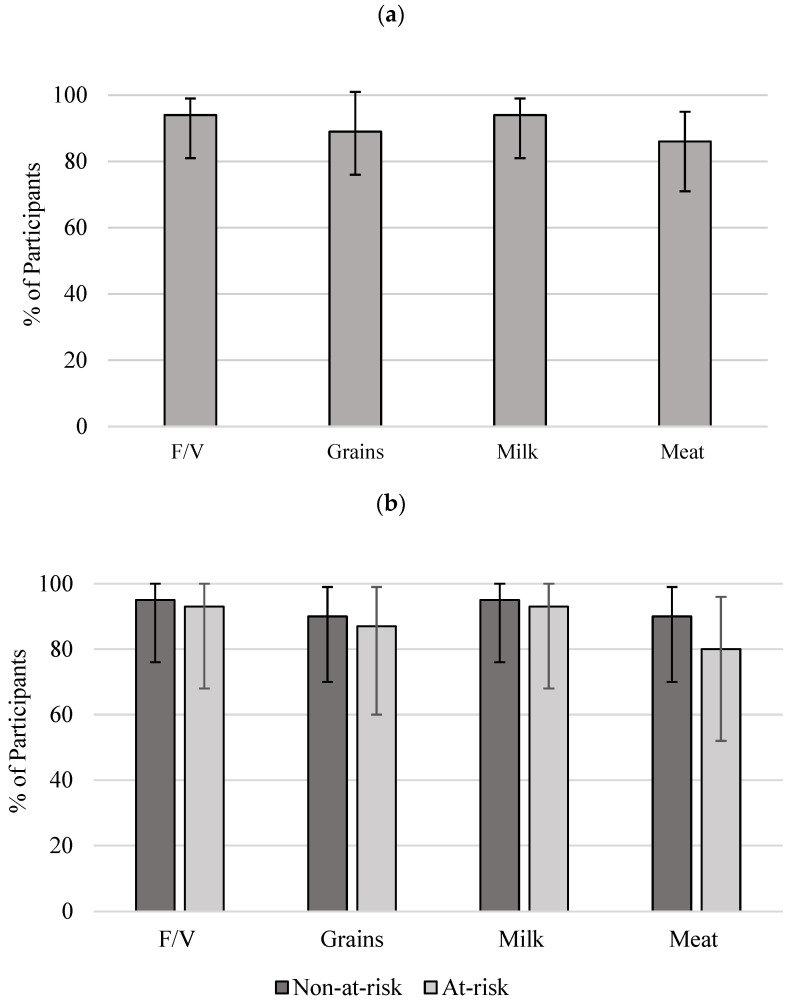
Proportion of participants and 95% CIs for not meeting CFG recommendations: (**a**) all women (*n* = 37); (**b**) non-at-risk (*n* = 22) and at-risk (*n* = 15). No significant difference was identified between the non-risk and at-risk groups. Reference intakes were adopted from CFG (2007) [20]. F/V, Vegetable and Fruit; Grain, Grain Products; Milk, Milk and Alternatives; Meat, Meat and Alternatives [20]. CI, confidence interval; CFG, Canada’s Food Guide.

**Figure 2 nutrients-14-03233-f002:**
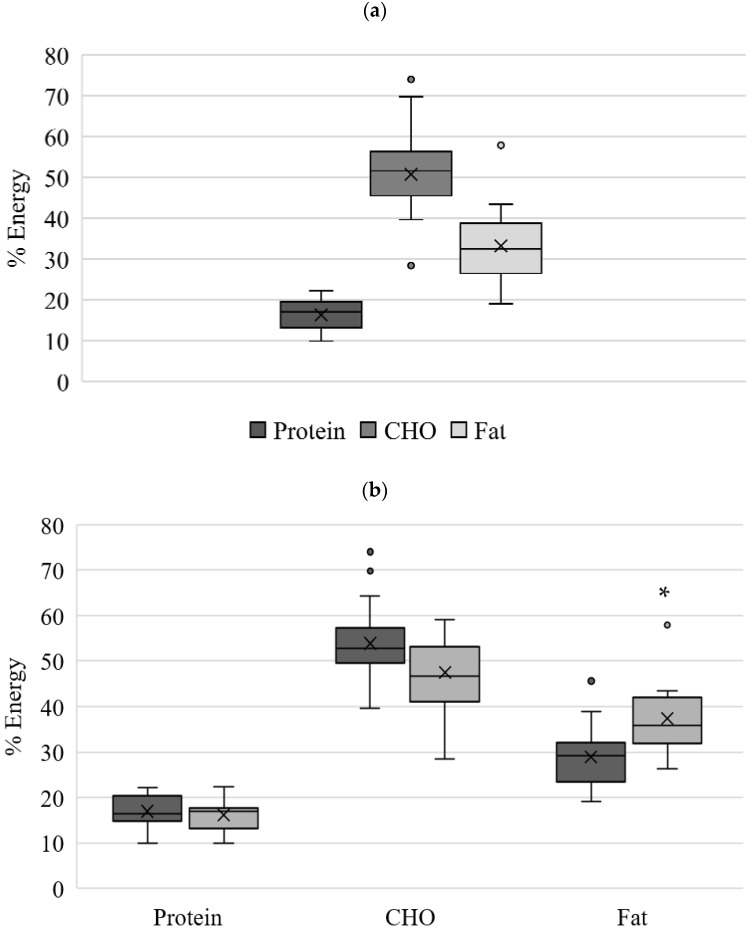
Energy intake (%) from macro-nutrients for (**a**) all women (*n* = 37); (**b**) non-at-risk (*n* = 22) and at-risk (*n* = 15). * significant difference (*p* < 0.05) between non-at-risk and at-risk groups. Total CHO includes added sugar. CHO, carbohydrates.

**Figure 3 nutrients-14-03233-f003:**
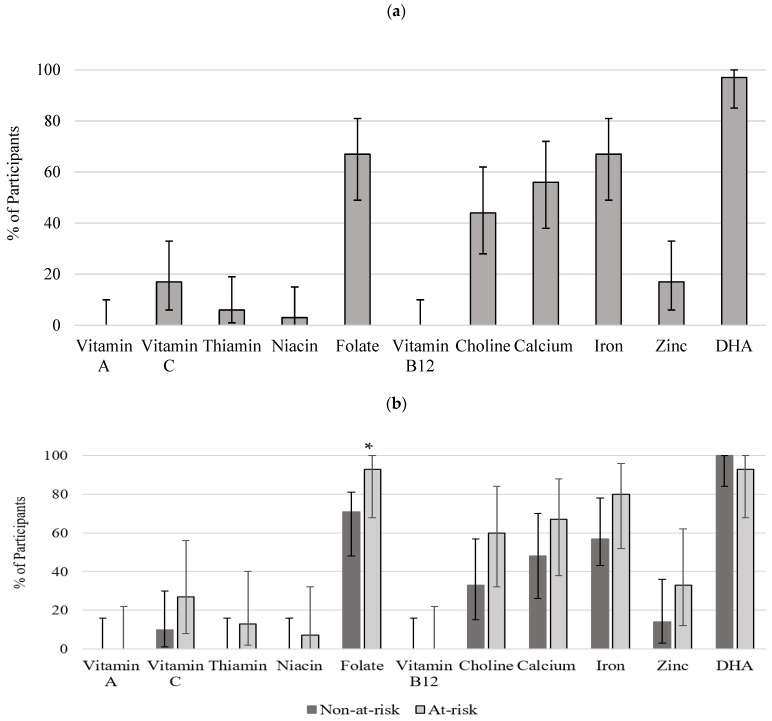
Proportion of participants not meeting DRI recommendations (**a**) all women (*n* = 37); (**b**) non-at-risk (*n* = 22) and at-risk (*n* = 15). Bars represent proportions and 95% CI. * significant difference (*p* < 0.05) between non-at-risk and at-risk groups. DRI: estimated average requirement (EAR) for pregnant women aged 14–18, 19–30, and 31–50 for vitamins A, C, B1, B12, folate, niacin, zinc, calcium, iron; adequate intake (AI) for choline [1]. Recommendations for DHA (C22:6n−3) were obtained from International Society for the Study of Fatty Acids and Lipids [24].

**Table 1 nutrients-14-03233-t001:** Maternal characteristics and health status for the participants and by alcohol use.

	All Women (*n* = 37)	Non-at-Risk (*n* = 22)	At-Risk (*n* = 15)	*p*-Value
Age ^a^ (Range)	24.4 ± 7.0 (14–42)	21.5 ± 5.5 (14–31)	28.1 ± 7.4 * (14–42)	0.007
Education ^b^ Elementary Junior high High school Post-secondary	1 (3) 16 (44) 15 (41) 5 (14)	1 (5) 12 (55) 6 (27) 3 (14)	0 5 (33) 8 (53) 2 (13)	0.196
Employment ^b^ Unemployed Employed part-time Employed full-time Student Maternity Leave	14 (39) 4 (11) 6 (16) 10 (28) 2 (6)	8 (38) 2 (9.5) 4 (19) 5 (24) 2 (9)	6 (40) 2 (13) 2 (13) 5 (33) 0 (0)	0.237
Social Assistance ^c^	19 (51)	10 (46)	9 (60)	0.385
Pre-pregnancy BMI ^a^ Below Normal Overweight Obese	26.5 ± 8.8 4 (12) 11 (33) 7 (21) 11 (33)	27.5 ± 8.3 2 (11) 6 (33) 3 (17) 7 (39)	25.9 ± 7.8 2 (13) 5 (33) 4 (27) 4 (27)	0.530
Chronic illness: Before pregnancy ^b^ During pregnancy ^b^	5 (15) 7 (21)	2 (10) 4 (20)	3 (20) 3 (20)	0.403 1.000
Smoking	11 (37)	7 (35)	6 (40)	0.762
Drugs ^c^	7 (19)	4 (19)	3 (20)	0.943
Pregnancy outcomes: # of pregnancies ^a^ # of miscarriages ^a^ # of stillbirths ^a^ # of full-term births ^a^ # of pre-term births ^a^	3.0 ± 2.2 0.6 ± 0.9 0.1 ± 0.4 1.0 ± 1.0 0.4 ± 1.0	2.7 ± 1.9 0.7 ± 1.0 0.1 ± 0.2 0.6 ± 1.4 0.3 ± 0.7	3.5 ± 2.6 0.5 ± 0.8 0.2 ± 0.4 1.4 ± 1.5 0.5 ± 1.6	0.439 0.620 0.190 0.110 0.690
Bed rest during pregnancy ^c^	6 (18)	5 (25)	1 (7)	0.179

Values are means ± SD and *n* (percentages). The differences between groups were tested by an Independent *t*-test ^a^, Wilcoxon rank-sum test ^b^ and a Chi-square or Fisher’s exact tests of independence ^c^. * significant difference between non-risk and at-risk group. BMI, body mass index; SD, standard deviation; #, numbers.

**Table 2 nutrients-14-03233-t002:** Food group intake of the participants and by alcohol use.

Food Group	Reference Intake	All Women (*n* = 37)	Non-at-Risk (*n* = 22)	Exposed (*n* = 15)	*p*-Value
Vegetable and Fruit	9	5 ± 3	5 ± 3	5 ± 3	0.783
Grain Products	8	5 ± 3	5 ± 3	5 ± 3	0.835
Milk and Alternatives	3	2 ± 1	2 ± 1	2 ± 2	0.570
Meat and Alternatives	2	3 ± 2	2 ± 2	3 ± 3	0.174

Values are means ± SD. Data derived from 24-h dietary recall. The differences between groups were tested by an Independent *t*-test. No significant difference was identified between non-risk and at-risk groups. Reference intakes were adopted from CFG (2007) [20].

**Table 3 nutrients-14-03233-t003:** Energy intake from each macronutrient for the participants and by alcohol use.

Macronutrient (Grams/Day)	All Women (*n* = 37)	Non-at-Risk (*n* = 22)	At-Risk (*n* = 15)	*p*-Value *
Protein	87 ± 35	82 ± 35	93 ± 35	0.390
CHO ^1^	257 ± 96	244 ± 105	272 ± 85	0.430
Fat	78 ± 49	58 ± 27	102 ± 59 *	0.010
Sugar	84 ± 46	89 ± 55	77 ± 33	0.510

Values are means ± SD. Data derived from 24-h dietary recall. The differences between groups were tested by an Independent *t*-test. * significant difference (*p* < 0.05) between non-at-risk and at-risk groups. ^1^ CHO, carbohydrates.

**Table 4 nutrients-14-03233-t004:** Daily micronutrient intake of the participants and by alcohol use.

		All Women (*n* = 37)		Non-at-Risk (*n* = 22)		At-Risk (*n* = 15)		*p*-Value
Micronutrient	Ref. DRI/Day	Intake	%DRI	Intake	%DRI	Intake	%DRI	
Vitamin A (RE)	550 (mcg)	1552 ± 1254	288 ± 229	1396 ± 613	258 ± 114	1815 ± 1833	339 ± 333	0.307
Vitamin C	70 (mg)	171 ± 128	251 ± 189	214 ± 152	313 ± 224	115 ± 53 *	172 ± 81	0.013
Thiamin (Vit B1)	1.2 (mg)	3 ± 3	272 ± 245	3 ± 1	267 ± 104	4 ± 4	294 ± 368	0.747
Niacin (Vit B2)	14 (mg)	34 ± 16	241 ± 116	39 ± 17	281 ± 123	29 ± 12 *	198 ± 80	0.029
Folate (Vit B9)	520 (mcg)	453 ± 194	86 ± 36	522 ± 200	70 ± 38	377 ± 142 *	10 ± 22	0.009
Vitamin B12	2.2 (mcg)	12 ± 10	540 ± 436	12 ± 6	565 ± 279	12 ± 13	535 ± 601	0.841
Choline	450 (mg)	524 ± 265	116 ± 59	601 ± 298	128 ± 65	441 ± 165	99 ± 46	0.123
Calcium	800 (mg)	1011 ± 531	116 ± 58	1126 ± 546	129 ± 62	885 ± 491	101 ± 47	0.153
Iron	22 (mg)	20 ± 10	88 ± 40	23 ± 9	101 ± 74	18 ± 9 *	74 ± 30	0.039
Zinc	9.5 (mg)	18 ± 10	176 ± 102	21 ± 12	206 ± 121	15 ± 8	142 ± 45	0.060
DHA	200 (mg)	78 ± 49	39 ± 25	93 ± 56	42 ± 24	79 ± 50	35 ± 26	0.383

Values are means ± SD. Data derived from FFQ. The differences between groups were tested by an Independent *t*-test. * significant difference (*p* < 0.05) between non-at-risk and at-risk groups. DRI: estimated average requirement (EAR) for pregnant women aged 14–18, 19–30, and 31–50 for vitamins A, B12, C, folate, vitamin B1, niacin, zinc, calcium, iron; adequate intake (AI) for choline. Recommendations for DHA (C22:6n−3) were obtained from International Society for the Study of Fatty Acids and Lipids [24]. FFQ, food frequency questionnaire; DRI, Dietary Reference Intakes; DHA, docosahexaenoic acid; RE, retinol equivalents; Vit, vitamin; Ref., reference.

## Data Availability

Will be available upon request with a permission of Opaskwayak Cree Nation Health Authority.

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
