# Peer review of "Dietary Intake Patterns and Lifestyle Behaviors of Pregnant Women Living in a Manitoba First Nations Community: Implications for Fetal Alcohol Spectrum Disorder"

_nutrients, 2022, doi:10.3390/nu14153233_

Round 1

Reviewer 1 Report

The manuscript did not change especially in its methodological section. It remains of small sample size and statistically underpowered.

Author Response

The manuscript did not change especially in its methodological section. It remains of small sample size and statistically underpowered.

 Response: Thank you for raising this point again. We are well aware of this limitation in our study. It was not easy to arrange in-person interviews with the pregnant mothers in this community. We have addressed the sample size issues and limitations in the previous revision. 

Reviewer 2 Report

Thanks for the possibility to review the manuscript titled “Dietary Intake Patterns and Lifestyle Behaviors of Pregnant 2 Women Living in a Manitoba First Nations Community: Implications for Fetal Alcohol Spectrum Disorder”.

I think that the manuscript is very important for the researchers and clinicians concerning the nutrition on maternal and fetal health and dietary patterns of First Nation women at-risk of FASD. The objectives are stated clearly and I think that this manuscript has the potential to be published.

The introduction has a clear and logical progression. The analysis of the literature provided is provides the basis for the study purposes. The text is well organized. The language is clear and the grammar is correct.

The sampling is appropriate (sample size).

The authors accurately explained how the data was collected and examined.  Furthermore, the authors clearly described the procedure.

Result section is well organized, the statistical analyses are detailed.
I think the discussion section did not have critical issues; authors explain their results and suggest hypotheses to interpret them. Moreover, the authors present strengths and limitations of the their study.

The findings presented in this study provide important baseline data on dietary patterns and nutrient intake of macro- and micronutrients important for fetal CNS development of pregnant women, especially of pregnant women at-risk of carrying a child with FASD.

In my opinion, it is an excellent work. I recommend the manuscript to publication and I have no relevant comments.

Author Response

Response to the reviewer 2:

Thanks for the possibility to review the manuscript titled “Dietary Intake Patterns and Lifestyle Behaviors of Pregnant 2 Women Living in a Manitoba First Nations Community: Implications for Fetal Alcohol Spectrum Disorder”.

I think that the manuscript is very important for the researchers and clinicians concerning the nutrition on maternal and fetal health and dietary patterns of First Nation women at-risk of FASD. The objectives are stated clearly and I think that this manuscript has the potential to be published.

The introduction has a clear and logical progression. The analysis of the literature provided is provides the basis for the study purposes. The text is well organized. The language is clear and the grammar is correct.

The sampling is appropriate (sample size).

The authors accurately explained how the data was collected and examined.  Furthermore, the authors clearly described the procedure.

Result section is well organized, the statistical analyses are detailed.
I think the discussion section did not have critical issues; authors explain their results and suggest hypotheses to interpret them. Moreover, the authors present strengths and limitations of the their study.

The findings presented in this study provide important baseline data on dietary patterns and nutrient intake of macro- and micronutrients important for fetal CNS development of pregnant women, especially of pregnant women at-risk of carrying a child with FASD.

In my opinion, it is an excellent work. I recommend the manuscript to publication and I have no relevant comments.

Response: thank you for the kind comments on our work. This study was extensive and difficult to complete with significant participants, as First Nations populations reside in remove and hard to reach areas. We fell this study is an important step in decolonizing Indigenous research and identifying data on all at -risk populations in Canada.

Reviewer 3 Report

Authors have written a manuscript entitled “Dietary Intake Patterns and Lifestyle Behaviors of Pregnant Women Living in a Manitoba First Nations Community: Implications for Fetal Alcohol Spectrum Disorder”. This was a pleasure to read and it is something that can benefit many HCPs. However improvement is needed. Please see my comments below. 

Abstract

Line 19: Write 37 in words at the start of the sentence.

Line 19: Add the word “in” after the word participate but before the word the.

Line 27: Write the percentage 68 in full at the start of the sentences

You can leave out the sentence with the non-significant result in the abstract.

Lines 32 and 33: Refine this sentence a little more and give examples of possible nutrition interventions and what type of benefit you are referring too.

Keywords

Put these in alphabetical order

Introduction

Authors have clearly shown the situation and the gap in literature. They have emphasized the need to conduct the current study.

Methodology

Was a pilot study conducted to test the FFQ? If so how was it modified to cater for the target population. If a pilot study was conducted who was used and how many?

What type of sampling was used to recruited the participants?

Did you validate your questionnaire by any experts in the field?

Results            

Please neaten up your table. You could bold significant p values for easy reference. Also create a space between the different categories so that the variables that are group together can easily be seen.

Please neaten table 3

Lines 243-245: Please check sentence for sense

Discussion

Lines 279-280: Do you mean normal weight or normal BMI. This sentence is confusing. Did your find 1 in 3 woman to have a normal BMI or to be overweight. Please rewrite sentence for clarity.

Why is been overweight and having co morbidities an issue in pregnancy. Give more details on this. What type of complications can arise? Pre-eclampsia? Gestational DM?

Will smoking effect the health of the baby in any way? Would it affect nutrition in any way?

You have done well in comparing the study results to other studies however you have missed out the “so what” factor in your discussion. You need to go a step further and reiterate what these study results mean for the health of the pregnant woman and what still needs to be done. You can strengthen your discussion to add greater depth and understanding.

Well done for including the study strengths and limitations.

Conclusion

It would be nice to include a way forward and also reiterate the so what factor. Maybe include a sentence or two on future studies.

Good Luck with the Revisions!

Author Response

Response to the reviewer 3:

Authors have written a manuscript entitled “Dietary Intake Patterns and Lifestyle Behaviors of Pregnant Women Living in a Manitoba First Nations Community: Implications for Fetal Alcohol Spectrum Disorder”. This was a pleasure to read and it is something that can benefit many HCPs. However improvement is needed. Please see my comments below. 

 Abstract

Line 19: Write 37 in words at the start of the sentence.

Response: revised as suggested.

Line 19: Add the word “in” after the word participate but before the word the.

Response: revised as suggested.

Line 27: Write the percentage 68 in full at the start of the sentences

Response: revised as suggested.

You can leave out the sentence with the non-significant result in the abstract.

Response: removed ‘no significant differences in pregnancy variables were detected between the two groups’ in the abstract.

Lines 32 and 33: Refine this sentence a little more and give examples of possible nutrition interventions and what type of benefit you are referring too.

Response: To accommodate this comment, the sentence has been revised to “The findings of this study may lay a fundamental premise for the development of community nutrition programs, nutrition education, and the foundation for nutrition intervention, such as community specific prenatal supplementation. These will assist in ensuring adequate maternal nutrient intake and benefit families and communities in Northern Manitoba with and without alcohol insult.”

Keywords

Put these in alphabetical order

Response: changed to be in alphabetical order as suggested.

Introduction

Authors have clearly shown the situation and the gap in literature. They have emphasized the need to conduct the current study.

Response: We value this feedback. Thank you!

Methodology

Was a pilot study conducted to test the FFQ? If so how was it modified to cater for the target population. If a pilot study was conducted who was used and how many?

Response: The study’s primary objectives do not include pre-testing the FFQ. Lack of proper testing (validation and reliability testing) is listed as the limitation. However prior to lunching the study in the community, content validity was undertaken in consultation with a number of community representatives, stakeholders, and professionals engaged with maternal programs in Winnipeg and First Nations communities across Manitoba (FASD community leaders, Mount Camel Clinic nurses and community liaison workers, Maternal and Child Health program experts.) This content was in the previous submission in the method.

What type of sampling was used to recruited the participants?

Response: We have recruited participants using convenience sampling. Individuals that attended the community’s “Prenatal Program” were invited to participate, those interested were screen for inclusion criteria and participated our study. This context has been updated in the method in this submission.

Did you validate your questionnaire by any experts in the field?

Response: The questionnaire was not validated. This has been addressed in the limitation section in the previous submission.

Results            

Please neaten up your table. You could bold significant p values for easy reference. Also create a space between the different categories so that the variables that are group together can easily be seen.

Please neaten table 3

Response: bolded significant p values and create a space between the different categories in Table 1 and improved table 3 to line up better (***no track changed were used for these changes due to busy looks)

Lines 243-245: Please check sentence for sense

Response: The revision has been made clearer by including the comparison group. Also found a spell mistake. Now the revision is in the lines 247-250.

Discussion

Lines 279-280: Do you mean normal weight or normal BMI. This sentence is confusing. Did your find 1 in 3 women to have a normal BMI or to be overweight. Please rewrite sentence for clarity. Response: This sentence has been improved to be clearer: “Maternal pre-pregnancy BMI was estimated to be overweight (26.5 ± 8.8 kg/ m2) with 33% being in normal BMI ranges.”

Why is been overweight and having co morbidities an issue in pregnancy. Give more details on this. What type of complications can arise? Pre-eclampsia? Gestational DM?

Response: This point has been addressed and information on complications and its outcomes has been added.

Will smoking effect the health of the baby in any way? Would it affect nutrition in any way? Response: Information on complications and its outcomes has been added.

You have done well in comparing the study results to other studies however you have missed out the “so what” factor in your discussion. You need to go a step further and reiterate what these study results mean for the health of the pregnant woman and what still needs to be done. You can strengthen your discussion to add greater depth and understanding.

Response: We have added so what factor in the discussion in lines of 339-408.

Well done for including the study strengths and limitations.

Response: thank you for this comment!

Conclusion

It would be nice to include a way forward and also reiterate the so what factor. Maybe include a sentence or two on future studies.

Response: Thank you for this point. An additional sentence on the importance for the findings has been added.

Good Luck with the Revisions!

Round 2

Reviewer 1 Report

Non 

Reviewer 3 Report

Authors have submitted a revised manuscript entitled “Dietary Intake Patterns and Lifestyle Behaviors of Pregnant Women Living in a Manitoba First Nations Community: Implications for Fetal Alcohol Spectrum Disorder”. This was a pleasure to read and is something that can benefit health care professionals and possibly contribute to the implementation of better measures during maternal care. Authors have provided detail comments on the changes made and have implemented the suggestions and comments given. This manuscript has significantly improved. There are some minor comments that need to be implement for further improvement. 

Line 101: Remove the word “the” before convenience sampling  

Line 283: replace “in” with the word “within”

Lines 292-293: change part of the sentence to read:

“Increasing number of reports indicate that metabolic conditions and diets that lack appropriate nutrients during pregnancy…

This manuscript is a resubmission of an earlier submission. The following is a list of the peer review reports and author responses from that submission.

Round 1

Reviewer 1 Report

Authors describes the dietary intake and lifestyle patterns of pregnant women in the Manitoba First Nations Community. They further compare the dietary intake of these pregnant women separated in groups as those consuming alcohol (at-risk) and those who did not consume alcohol (non-at-risk) during pregnancy. Even though the sample size is small when separated for at-risk and non-at-risk groups with respect to alcohol, the data is analyzed appropriately. The strengths and limitations of the analysis is also well addressed in the manuscript. In conclusion, the authors describe the lower intake of important nutrients such as iron, folate and DHA during pregnancy in the at-risk women, which are essential for fetal CNS development.

The figures, however, needs to be revised to ensure the error bars are not cut off and the asterisk (*) exhibiting significant difference is visible.

Title of manuscript: miss-spelled “patterns” as “patters”

Results- basic demographics-line 1: characteristics “of” women…missing of

Line 3: Rewrite the sentence “Thirty-eight (38) women participated in this study and one incomplete” for clarity.

Food group intake line 4: check and correct “milk and alternative” written twice.

Table 3: Add full form of CHO in footnote

Figure 2b: add legend

Author Response

Response to the Academic Editor:

Please define “at risk” and “not at risk” early in the abstract and methods

Response:

This has been defined in the abstract and early in the method section as suggested.

 Response to the Reviewer 1:

 Authors describes the dietary intake and lifestyle patterns of pregnant women in the Manitoba First Nations Community. They further compare the dietary intake of these pregnant women separated in groups as those consuming alcohol (at-risk) and those who did not consume alcohol (non-at-risk) during pregnancy. Even though the sample size is small when separated for at-risk and non-at-risk groups with respect to alcohol, the data is analyzed appropriately. The strengths and limitations of the analysis is also well addressed in the manuscript. In conclusion, the authors describe the lower intake of important nutrients such as iron, folate and DHA during pregnancy in the at-risk women, which are essential for fetal CNS development.

  1. The figures, however, needs to be revised to ensure the error bars are not cut off and the asterisk (*) exhibiting significant difference is visible.

  Response:  This has been addressed through the re-formatting of the figures 1 and 3.  Error bars represent 95% confidence interval error.

 Title of manuscript: miss-spelled “patterns” as “patters”

Response: This has been revised.

  1. Results- basic demographics-line 1: characteristics “of” women…missing of

Response: This has been corrected

  1. Line 3: Rewrite the sentence “Thirty-eight (38) women participated in this study and one incomplete” for clarity.

Response: Although 38 women participated in this study, but only 37 were included for data analysis since one participant did not provide dietary records.  This line has been updated in this submission.

  1. Food group intake line 4: check and correct “milk and alternative” written twice.

Response: Thank you for pointing this out. It has been corrected

  1. Table 3: Add full form of CHO in footnote

Response: The footnote has been added under all of the figures.

  1. Figure 2b: add legend

Response: It has been added in the legend in the original submission.

Reviewer 2 Report

Major comment 

The major concern about this study is the small sample size. This feature cannot be underestimated, since it affects also the generalization of the finding. Moreover the statistical analysis can be arguable, for instance in multivariable and multi comparison analysis, the adjustment of statistical significance is needed. On the other hand the statistical analysis in underpowered, where larger sample is needed to detect statistical significance.        

Other comments 

The title is unclear and to be rephrased.  The abstract is too long and should be shortened not to exceed 200 words and formatted properly with no subheadings.  

Author Response

Response to the Reviewer 2.

  1. Major comment 

The major concern about this study is the small sample size. This feature cannot be underestimated, since it affects also the generalization of the finding. Moreover the statistical analysis can be arguable, for instance in multivariable and multi comparison analysis, the adjustment of statistical significance is needed. On the other hand the statistical analysis in underpowered, where larger sample is needed to detect statistical significance.        

Response: Thank you for pointing this out. In the original submission, the authors clearly stated this small sample size issues in the limitations section of the paper. Due to the data being collected from hard-to-reach, transient First Nations pregnant population, this sample size is considered acceptable as convenient samples. The statistical analysis did not include multivariable analysis. The statistical analysis of this study was completed in consultation with a statistician who provided guidance since the inception of the study. The authors agree that the analysis may present itself as underpowered, however due to lack of nutrition data on First Nations population at-risk of carrying a child with FADS, the results from this pilot may be used as a baseline for future research with similar cross-section design.

  1. Other comments 

The title is unclear and to be rephrased.  The abstract is too long and should be shortened not to exceed 200 words and formatted properly with no subheadings.

Response: The title has been changed to “Determining the implications for fetal alcohol spectrum disorder on dietary intake patterns and life styles of pregnant women living in a Manitoba First Nations community”. The abstract has been substantially shortened, from 400 to 258 words.

Round 2

Reviewer 2 Report

The methodological flaws persist. 

Author Response

  1. The methodological flaws persist

Response:

Thank you again for pointing this out.

In the previous response and also in the original submission, we admit the small sample size issues. Due to the data being collected from hard-to-reach, a long distance but in person meeting with the participants, and transient First Nations pregnant population, this sample size is considered acceptable as convenient samples. The statistical analysis did not include multivariable analysis. The statistical analysis of this study was completed in consultation with a statistician who provided guidance since the inception of the study. The authors agree that the analysis may present itself as underpowered, however due to lack of nutrition data on First Nations population at-risk of carrying a child with FADS, the results from this pilot may be used as a baseline for future research with similar cross-section design.